# A Novel Caffeine Oleate Formulation as an Insecticide to Control Coffee Berry Borer, *Hypothenemus hampei*, and Other Coffee Pests

Carmenza E. Góngora [1,*], Johanna Tapias [1], Jorge Jaramillo [1], Rubén Medina [2], Sebastián González [3], Tatiana Restrepo [3], Herley Casanova [3] and Pablo Benavides [1]

1   Department of Entomology, National Coffee Research Center, Cenicafé, Manizales 170009, Colombia; johatapias@gmail.com (J.T.); jorlu7@gmail.com (J.J.); pablo.benavides@cafedecolombia.com (P.B.)
2   Department of Biometrics, National Coffee Research Center, Cenicafé, Manizales 170009, Colombia; ruben.medinal@cafedecolombia.com
3   Colloids Group, Institute of Chemistry, University of Antioquia, Medellín 050010, Colombia; asgonzalezgh@gmail.com (S.G.); tatianacoloides@gmail.com (T.R.); herley.casanova@udea.edu.co (H.C.)
*   Correspondence: carmenza.gongora@cafedecolombia.com; Tel.: +57-3154869466

**Abstract:** The coffee berry borer (CBB), *Hypothenemus hampei* (Ferrari, 1867) (Coleoptera: Curculionidae: Scolytinae), is the pest that causes the most economic damage to coffee crops. Chemical control of this insect is based on the use of insecticides that can affect the environment and nontarget organisms. Despite the fact that caffeine has shown potential as an insecticide, a caffeine-based product for field use is currently not available on the market. As a new alternative to control CBB and other coffee pests, such as *Monalonion velezangeli* Carvalho and Costa, 1988 (Hemiptera: Miridae), a caffeine-oleate was developed. The caffeine oleate formulation showed laboratory efficacy by causing mortality of more than 90% of CBB adults in preventive tests in which the insecticide was sprayed prior to insect attack on the coffee fruits. In the curative tests, in which spraying occurred after CBB infested the fruits, the formulation caused 77% mortality of the insects. Under controlled field conditions, the product kept CBB infestation below 20%, reducing the number of fruits attacked by the insect by up to 70%. In addition, no phytotoxic effects were observed in coffee plants. The insecticide was also effective against *M. velezangeli*, causing 100% mortality. A caffeine oleate formulation that could be part of a strategy for integrated CBB management as well as other pests of coffee was developed. The components of the insecticide are food grade, and the product would provide greater security to the coffee ecosystem and coffee growers.

**Keywords:** coffee berry borer; caffeine; oleic acid; kaolin; new insecticide; infestation

## 1. Introduction

The coffee berry borer (CBB), *Hypothenemus hampei* (Ferrari, 1867) (Coleoptera: Curculionidae: Scolytidae), native to equatorial Africa, is found in all coffee-growing regions of the world [1,2]. This insect causes great economic losses and is difficult to manage because its life cycle takes place within the coffee fruit [3–5]. Currently, for the chemical control of the insect, insecticides of two groups are recommended: organophosphates and anthranilic diamides [6,7]. These products are of toxicological category III (World Health Organization WHO classification), and although they are effective [8,9], their toxicity in the environment, the resistance that can be generated in other insects and their current status in global agricultural production make them susceptible to being restricted in the future, which is why it is necessary to expand pesticide options for pest control [10].

*Monalonion velezangeli* Carvalho and Costa, 1988 (Hemiptera: Miridae) is known as the avocado bug or coffee bug [11,12]. It is a polyphagous insect that attacks vegetative shoots, flowers, inflorescences, and fruits of ornamental plant species, weeds, and timber and fruit

trees, including coffee [12–14]. This pest causes damage by puncturing the vegetable and sucking the sap, in turn necrotizing the tissues. Strategies for pest management do not yet exist, and some producers from some Colombian avocado-coffee zones use insecticides without prior evaluation of their effects [12].

Phytochemical compounds derived from plants that are capable of synthesizing secondary metabolites with biological properties against insect pests and other microorganisms have been proposed as new alternatives for the development of agrochemicals [13–17]. The compounds that have been isolated are as varied as the plants from which they have been extracted, and their protective effects range from repellency and deterrence of feeding and oviposition to acute toxicity and interference with the growth and development of insects [18,19].

One of the most recognized phytochemicals with a potential insecticide effect is caffeine (1,3,7-trimethylxanthine), due to its repellency properties [20,21] and its effects on the central nervous system by inhibiting $Ca^{2+}$ transport [22–26]. In insect neurons and muscles, calcium channels are considered site-of-action targets for the development of pesticides, given their key role in multiple biological processes, including cell signaling and neurotransmitter release [27,28].

Caffeine is an alkaloid of the methylxanthine family. It is an odorless, colorless, and bitter powder that is used to manufacture analgesics as a potentiator or excitant of the central nervous system [29]. In veterinary medicine, it is used as a cardiac and respiratory stimulant, and, in the food industry, it is used to manufacture cola and energy drinks [30]. Generally, caffeine is classified as safe, and, according to the US Food and Drug Administration (FDA), it can be part of the healthy diet of most people [31].

Caffeine has been widely consumed by vertebrates for several centuries, because it is naturally produced in variable amounts in the seeds, leaves, and fruits of some edible plants, including coffee, tea, mate, and guarana [32]. Its production in these plants is associated with the theory of chemical defense, since it acts as a defense mechanism in response to herbivory such as insects and microorganisms [22]. The first direct evidence of caffeine acting as a plant chemical defense was proposed by Nathanson (1984), who showed that caffeine and other methylxanthines interfere with the feeding and reproduction of *Manduca sexta* on *Lycopersicon esculentum* [33]. In tea (*Camellia sinensis*), it has been proposed that the accumulation of caffeine in the stems after attack by adults of the beetle *Xyleborus fornicatus* could be a defense strategy of the plants, because it inhibits the growth of the fungus *Monacrosporium ambrosium*, a symbiote necessary for the development of *X. fornicates* [34]. Other studies have shown that this alkaloid also inhibits oviposition and slows the growth of *X. fornicates* [35]. Additionally, the growth of the fungus *Crinipellis perniciosa* in cocoa is significantly inhibited by caffeine, and the stem tissue infected by the fungus can contain between seven and eight times more caffeine than healthy stem tissue [36].

Spraying tomato leaves with a 1% caffeine solution decreases food intake by *M. sexta*, while treating cabbage leaves and orchids with 0.01 to 0.1% caffeine solutions causes neurotoxic reactions and kills or repels slugs and snails [37]. Caffeine at a concentration of 200 µg/mL blocks the development of *Aedes aegypti* (Diptera, Culicidae) in the larval phase and inhibits molting in adults, with no evidence of resistance in subsequent generations after caffeine treatment [38]. Additionally, caffeine has been shown to inhibit the growth of mites [39].

Despite the fact that caffeine has shown potential as an insecticide in many species, a caffeine-based product for field use is currently not available on the market. This might be related to its low solubility on both aqueous and organic phases that limits formulation payload. Moreover, caffeine does not easily penetrate the chitinous exoskeleton of insects [40], and some insects have detoxification mechanisms at the level of the digestive system [41–43].

The role of caffeine in the chemical defense of plants against CBB has been investigated. There is no positive correlation between the caffeine content in the seeds of the coffee

plant and the resistance of the plant to the insect. These results indicate that *H. hampei* has developed an adaptation to manage the toxic effects of caffeine ingestion [44]. This adaptation of *H. hampei* is mainly mediated by the activity of its intestinal microbiota, which is responsible for degrading caffeine in the insect's digestive tract [45]. However, Araque and collaborators (2007) determined the bioactivity of aqueous solutions of caffeine (0.20–2.00%) and emulsions of caffeine oleate (20% oil, 2% surfactant, 0.04% caffeine, and 0.05% oleic acid) in contact with the cuticle of insects, including *H. hampei* and *Drosophila melanogaster*. The aqueous solutions did not show insecticidal activity, while the caffeine oleate emulsions did, with a $TL_{50}$ of approximately 17 min observed for *H. hampei*. This study demonstrated the effect of caffeine on CBB when formulated as an oil-in-water (O/W) emulsion [40]. Caffeine appears to cause toxicity to the insect after being conveyed in the oil phase of the emulsion. The results showing the biological activity of the caffeine oleate emulsion against *H. hampei* indicated the possibility of using such an emulsion as a topical insecticide for the control of this pest. Additionally, some studies have reported on the insecticidal properties and mechanisms of action of mineral particles such as kaolin due to the abrasive and adsorptive properties of this mineral on insect cuticles [46,47].

With the hypothesis that a suitable caffeine formulation can have an insecticidal effect by causing the compound to penetrate the cuticle of insects and act at the receptor level and given the need to evaluate new substances with low environmental impact, a caffeine-based formulation with oleic acid, Tween 80, and kaolin was developed, and its efficacy in the control of CBB and other coffee pests was determined.

## 2. Materials and Methods

### 2.1. Caffeine Oleate Formulation

This study was carried out in the Entomology Laboratory of the National Coffee Research Center (Cenicafé), located in Manizales (Caldas, Colombia) at 04°59′ N latitude, 75°35′ W longitude, and an altitude of 1413 m.

After evaluating different concentrations of oleic acid and caffeine, an emulsifiable concentrate of caffeine oleate was prepared by mixing oleic acid (6 g), Tween 80 (0.3 g), and caffeine (0.4 g). The mixture was stirred on a heating plate (PC-620D, Corning, New York, NY, USA) at a temperature of 80 °C for 10 min. The mixture was then stirred until it cooled to room temperature.

A caffeine oleate emulsion was prepared from the emulsifiable concentrate by mixing 6.66 g concentrate with 93.4 g water. In addition, another emulsion was prepared, to which functionalized kaolin (0.1 g, Nexentia-Sumicol S.A., Medellín, Colombia) was added. The biological efficacy of the two caffeine oleate emulsions against CBB was determined by evaluating the effect of the product applied to coffee fruits before (preventive effect) and after (curative effect) infestation by *H. hampei*.

### 2.2. Preventive Effect of Caffeine Oleate on CBB under Laboratory Conditions

Coffee fruits, with an age of 120 to 150 days of development and containing more than 20% dry matter, were collected from *Coffea arabica* variety Castillo® trees and disinfected with 0.5% sodium hypochlorite and ultraviolet (UV) light for 15 min [48]. Once disinfected, the fruits were sprayed with the emulsions or treatments identified in Table 1. The spray applications were carried out with a portable sprayer (Portable Spray Gun System, Preval®, Bridgeview, IL, USA), and in all the cases the volume application was of 500 µL. As a control, distilled water was applied.

**Table 1.** Treatment descriptions.

| Code | Treatment | Caffeine | Oleic Acid | Kaolin |
|------|-----------|----------|------------|--------|
| T1 | Control (Water) | | | |
| T2 | Emulsion | 4000 µg mL$^{-1}$ | 6% (p/p) | |
| T3 | Emulsion + kaolin | 4000 µg mL$^{-1}$ | 6% (p/p) | 0.1% |

For the application of the treatments, 5 groups of 30 fruits were sprinkled with the corresponding treatment and placed in $16.5 \times 11 \times 7$ cm methacrylate boxes with lids. Sixty insects were added to each group, forming the experimental units (Figure 1). Five repetitions were used per treatment.

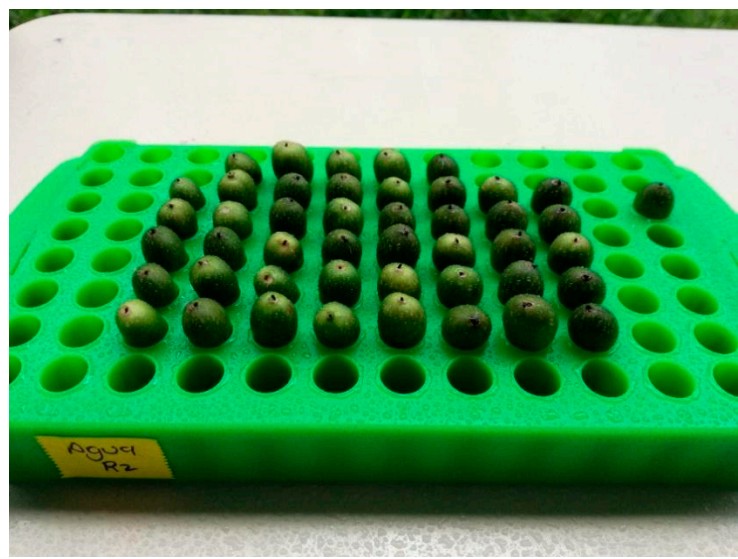

**Figure 1.** Experimental unit for the evaluation of the postinfestation (curative) effect. A 96-well rack with 46 fruits infested by CBB.

After spraying, the fruits were kept in the laboratory in a laminar flow chamber (AVC-4D2, Esco, Horsham, PA, USA) for one hour to dry them. Subsequently, they were infested by adding newly emerged CBB adult females provided by the breeding laboratory (Biocafé, Chinchiná Colombia). The CBB were previously disinfected by immersion in a 0.5% sodium hypochlorite solution [49].

### 2.3. Curative Effect of Caffeine Oleate on CBB under Laboratory Conditions

To evaluate curative effects, coffee fruits were disinfected [48]. Groups of 90 fruits were placed in $16.5 \times 11 \times 7$ cm methacrylate boxes with a lid, and 120 disinfected adult insects were added to these boxes [49].

Infestation was achieved after 2 h. Then, the infested fruits with CBB in positions A and B in the fruit pericarp but that had not yet bored into the coffee seed [48,49] were selected, 46 infested fruits were chosen and placed in a 96-hole plastic rack (HD23461D, Heathrow Scientific, Vernon Hills, IL, USA) (Figure 1), and the treatments in Table 1 were applied to these fruits. The treatments were applied directly to CBB in positions A and B. For the applications, a Preval® sprinkler was used, and in all the cases the volume application was of 500 μL. Each treatment was applied to 3 racks, representing experimental units. Three repetitions were used per treatment.

For both preventive and curative tests, the experimental units were stored under controlled conditions: a temperature of $25 \pm 2$ °C, a relative humidity of $71 \pm 5$%, and a photoperiod of 12 h. The fruits were examined after 1, 5, 10, and 20 days. The number of dead CBB and number of infested fruits were quantified.

The response variables were the percentage of mortality over time and the percentage of healthy seeds. Data were expressed as means and Sd (standard deviation) for each one of the variables. The effect of treatments was analyzed using ANOVA according to the model for a complete randomized design at 5% and statistical significance of $p < 0.05$.

Dunnett's 5% comparison test was performed to compare the treatments with the absolute control (water control).

Since the treatments were significantly different from the control, a 5% least significant difference (LSD) test was applied to compare the two emulsions. In addition, the power of

the test was evaluated, which was greater than 85%, and the assumptions of normality and homogeneity of variances were met.

### 2.4. Effect of Caffeine Oleate Emulsion Components against CBB under Laboratory Conditions

To understand the mode of action of the caffeine oleate emulsion and to know the effect of each one of the components, the pre-infestation and post-infestation effect of each emulsion component was evaluated (Table 2). In T5, the mixture of oleic acid 6% and caffeine 4000 µg mL$^{-1}$ was heated at a temperature of 80 °C for 10 min, in order to obtain the caffeine oleate emulsion. In T4 the mixture was not heated.

**Table 2.** Treatment descriptions.

| Treatment | Caffeine | Oleic Acid | Tween | Kaolin |
|---|---|---|---|---|
| T1. Control (Water) | | | | |
| T2. Oleic acid 6% | - | 6% (p/p) | 0.3% | - |
| T3. Caffeine 4000 µg mL$^{-1}$ | 4000 µg mL$^{-1}$ | 6% (p/p) | 0.3% | - |
| T4. Oleic Acid 6% + Caffeine 4000 µg mL$^{-1}$ | 4000 µg mL$^{-1}$ | 6% (p/p) | 0.3% | - |
| T5. Caffeine oleate emulsion (Emulsion + kaolin) | 4000 µg mL$^{-1}$ | 6% (p/p) | 0.3% | 0.1% |

The response variables were the percentage of mortality over time and the percentage of healthy seeds. Data were expressed as means and Sd (standard deviation) for each one of the variables. The effect of treatments was analyzed using ANOVA according to the model for a complete randomized design at 5% and statistical significance of $p < 0.05$. Tukey's 5% comparison test was performed to compare the treatments.

### 2.5. Evaluation of the Preventive Effect of Caffeine Oleate on CBB in the Field

The assay was carried out on a commercial coffee plantation of three-year-old Castillo® variety coffee trees located at the "La Catalina" Cenicafé experimental station in the municipality of Pereira (Risaralda, Colombia) at 04°58′ N latitude, 75°42′ W longitude, and a 1400 m altitude. The plantation had 2000 trees, in which two plots were selected. In each plot, 25 trees with a productive branch bearing fruits at more than 120 days of development were chosen at random. The previously infested and ripe fruits on the selected branches were removed, leaving approximately 50 healthy fruits on each branch.

One treatment was applied to each of the 25 selected branches (repetitions) in each plot: T1, distilled water as a control, and T2, caffeine oleate emulsion with kaolin. For the application of the emulsion, 200 cc was prepared, weighing the emulsifiable concentrate of caffeine oleate (13.6 g), functionalized kaolin (0.2 g), and water (186.6 g).

The spray applications were carried out with a manual sprayer (Royal Condor®, Soacha, Cundinamarca, Colombia) with a TXVK-3 nozzle, applying 5 cc to each branch. One hour after application, the branches of all treatments were covered with entomological sleeves, and 100 newly emerged adult CBB were released into each sleeve. Each experimental unit consisted of one branch with 50 coffee fruits infested by CBB.

One day and 15 days after application, the number of CBB-infested fruits on each selected branch were counted. Additionally, the number of dead CBB in the sleeves were counted at 1, 3, 5, 10, and 15 days after application. On day 15 all the fruits on each branch were collected and taken to the laboratory to quantify the total number of infested fruits per branch and identify the position of penetration of CBB. With this information, the percentage of infestation, percentage of insect mortality over time, and percentage of seeds damaged by the insect were determined. Averages with their respective confidence intervals were calculated.

### 2.6. Evaluation of the Curative Effect of Caffeine Oleate on CBB in the Field

The curative effect of the emulsion of caffeine oleate with kaolin was assessed following the methodology described for evaluating the preventive effect and in the same plots after completing the preventive effect test but on another group of trees.

The treatments applied were the same as those in the previous experiment: T1, distilled water as a control, and T3, caffeine oleate emulsion with kaolin. However, prior to the application of the treatments, the branches were covered with entomological sleeves, and 100 adult insects were released inside each one. Twenty-four hours after infestation by CBB, the insects that were not infested were removed from the sleeves, and the branches with the fruits infested by CBB were sprayed with the respective treatments. After the application, the branches were left covered by the sleeves to avoid infestations external to the bioassay.

The number of fruits infested by CBB on each branch and the number of dead CBB in the sleeves were counted 1, 2, 5, and 15 days after application. After 15 days, all the fruits on each branch were collected and taken to the laboratory. Analyses was done as described previously for 2.5.

### 2.7. Evaluation of the Curative Effect of Caffeine Oleate on M. velezangeli under Laboratory Conditions

Adults and nymphs of *M. velezangeli* were collected in the field from coffee plants on farms in the El Billar village in Ansermanuevo, Valle del Cauca (4°48′57.1″ N latitude and 76°05′40.2″ W latitude) at an altitude of 1849 masl. A caffeine oleate emulsion with kaolin was prepared from the emulsifiable concentrate of caffeine (6.6 g) mixed with water (93.3 g) and kaolin (0.1 g). The treatments consisted of the application of T1 (distilled water as a control) and T3 (the caffeine oleate emulsion with kaolin) (Table 1).

Sixty insects, nymphs and adults of *M. velezangeli*, were separated into groups of three insects, which were placed in 7.5 cm tall × 21.5 cm long × 10 cm wide rectangular boxes containing a wet napkin at the base and *Cissus verticillata* stems with leaves. Each box was an experimental unit (Figure 2). The treatments were sprayed with a Preval® spray sprinkler, with application to the insects and the substrate. Ten experimental units with three insects (thirty insects total) per treatment were evaluated. After spraying, the boxes were incubated under controlled conditions at a 20 °C temperature, an 80–90% relative humidity, and a 12 h photoperiod.

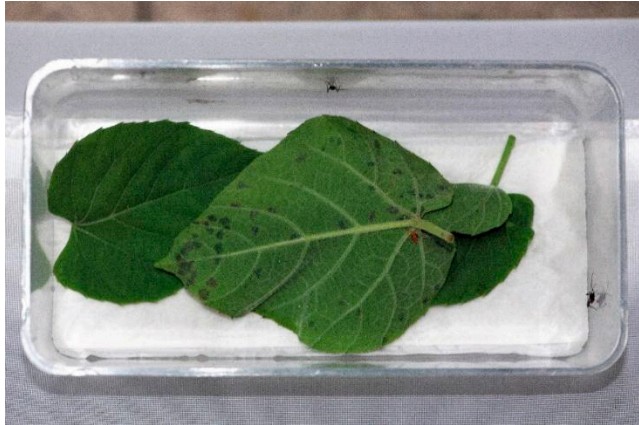

**Figure 2.** Experimental unit for the evaluation of the emulsion effect on *M. velezangeli*. *C. verticillata* plant arranged on a napkin with two adults and a nymph.

For 12 days, the insects were checked daily, counting the number of dead individuals and the number of nymphs that entered the adult stage. The evaluation days were determined by the state of the plant material in the control (T1) treatment.

The response variable was the percentage of insect mortality after 12 days. The mortality curve over time and the proportion of insects that reached the adult stage were

determined. An analysis of variance was performed to determine differences in mortality between the control (T1) and caffeine emulsion treatment (T2).

### 2.8. Evaluation of Anatomical Changes of CBB in Contact with Caffeine Oleate by Using Scanning Electron Microscopy

CBB adult females, 30 insects provided by the breeding laboratory (Biocafé, Chinchiná, Colombia), were assigned to three groups with 10 insects per group, the first group corresponding to the non-treated insects, the second group to the insects that were sprayed with 300 uL of the caffeine oleate emulsion including kaolin (COEK), and the last group to the insects that were in contact with the insecticide by walking on a piece of towel paper that cover the base of a petri dish, wetted with 400 uL of (COEK). A morphological analysis of the three group of insects was carried out by using a JSM 6490 LV scanning electron microscope (JEOL, Akishima, Japan). All insects from each group were fixed on a metallic coin by a conductive carbon tape. Afterwards, insect protein fixation was carried out by immersing the specimen in a glutaraldehyde bath followed by a dehydration process using ethanol and, finally, the deposition of a gold coat using a sputter coating system Desk IV (Denton Vacuum, Moorestown, NJ, USA). All samples were analyzed at a work distance of 5 mm and a voltage of 15.0 kV.

## 3. Results

### 3.1. Preventive Effect of Caffeine Oleate on CBB under Laboratory Conditions

In this experiment, the effect of the emulsions on CBB when applied to the coffee fruits was observed after 24 h, and the mortality increased over the observed days (Figure 3). The two emulsions (Table 1) caused high insect mortality (84%) on day 20 (Table 3) and showed significant differences compared to the control with water (ANOVA, F = 453.48, DF = 2.24, $p < 0.0001$). According to the LSD test, the addition of kaolin potentiated the effect of the emulsion, resulting in a higher (94%) mortality ($p < 0.001$) in this treatment.

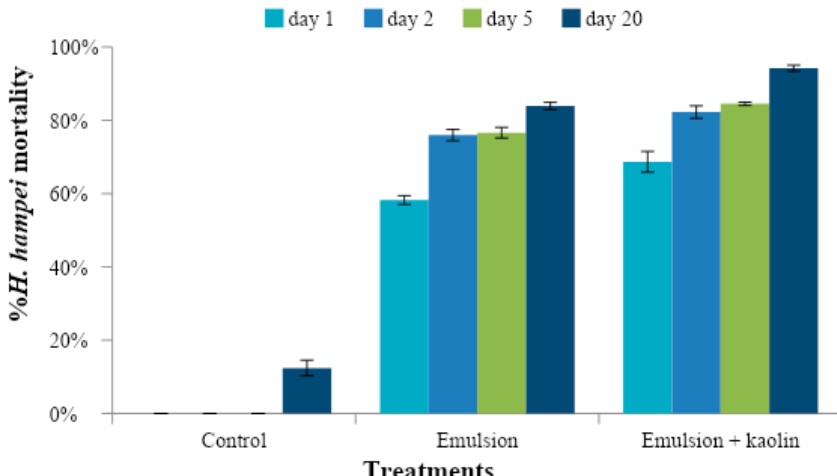

**Figure 3.** Percentage of CBB mortality evaluated on day 1, 2, 5 and 20 after preinfestation treatment.

**Table 3.** Percentage of CBB mortality and of healthy seeds at 20 days after preinfestation treatment.

| Treatment | Experimental Unit | Mortality (%) | | Healthy Seed (%) | |
|---|---|---|---|---|---|
| | | Average | Sd | Average | Sd |
| Control (water) | 15 | 12.4 | 8.3 | 37.0 | 6.3 |
| Emulsion | 15 | 83.9 *b | 3.9 | 89.0 *b | 6.0 |
| Emulsion + kaolin | 15 | 94.2 *a | 3.2 | 9.42 *a | 3.7 |

* For each variable, different letters indicate differences with respect to the control (water) according to Dunnett's test at 5%. Sd = Standard deviation.

Regarding the percentage of healthy seeds (Table 3), there were also differences between the control and the emulsions (ANOVA, F = 76.28, DF = 2.24, *p* < 0.0001) and according to Dunnett's test at 5% (*p* < 0.001). In the control, 37% of the seeds were not infested, while with the application of the emulsions, protection of the seeds was observed, with 94% of the seeds remaining healthy with the emulsion containing kaolin and 89% with the emulsion alone.

### 3.2. Curative Effect of Caffeine Oleate on CBB under Laboratory Conditions

The applications of caffeine oleate emulsions (Table 1) to the infested fruits had an effect on CBB (Figure 4), and significant differences were found between the emulsions and the control (Table 4). The highest mortality occurred with the kaolin emulsion, which differed from the emulsion without kaolin (ANOVA, F = 947.34, DF = 2.42, *p* < 0.0001).

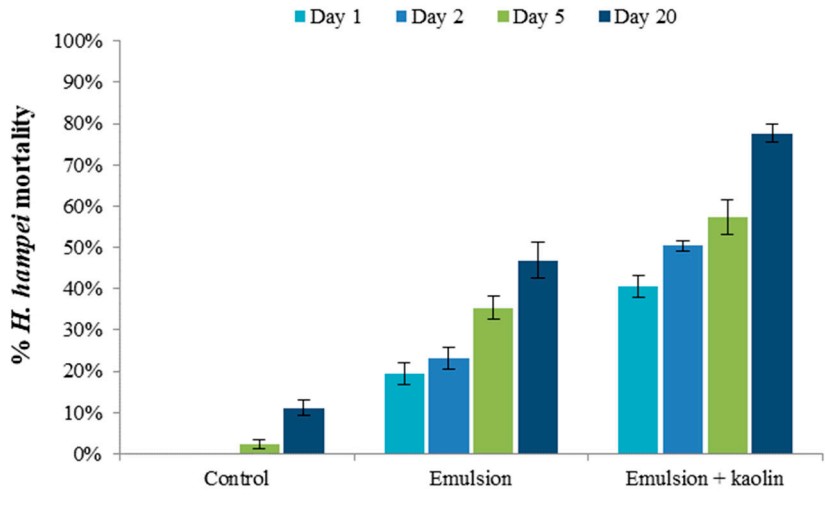

**Figure 4.** Percentage of CBB mortality evaluated on day 1, 5 and 20 after postinfestation treatment.

**Table 4.** Percentage of CBB mortality and of healthy seeds at 20 days after postinfestation treatment.

| Treatment | Experimental Unit | Mortality (%) | | Healthy Seed (%) | |
|---|---|---|---|---|---|
| | | Average | Sd | Average | Sd |
| Control (water) | 9 | 11.1 | 3.0 | 57.3 | 3.9 |
| Emulsion | 9 | 46.8 *b | 6.6 | 79.2 *b | 8.6 |
| Emulsion + kaolin | 9 | 77.8 *a | 3.7 | 90.0 *a | 2.9 |

* For each variable, different letters indicate differences with respect to the control (water) according to Dunnett's test at 5%. Sd = Standard deviation.

The effect of the emulsions was reflected in the percentage of healthy seeds on day 20 of evaluation (Table 3), since, due to high insect mortality, the insect did not manage to penetrate the coffee seeds and damage them. Application of the emulsions protected between 79 and 90% of the coffee seeds, showing differences with respect to the control, in which 57% of the seeds were found healthy (ANOVA, F = 502.54, DF = 2.42, *p* < 0.0001). Significant differences were also observed between the two emulsions in favor of the emulsion containing kaolin (*p* < 0.01).

### 3.3. Effect of Caffeine Oleate Emulsion Components against CBB under Laboratory Conditions

In the preventive effect evaluation of treatment from (Table 2), the mortality recorded on the control was less than 10% (Figure 5). At day 20 no differences were found among the control, oleic acid, and caffeine (ANOVA, F = 498.78, DF = 4.20, *p* < 0.0001). The mortalities caused by oleic acid plus caffeine were below 60%. Meanwhile the caffeine oleate emulsion

showed a higher effect causing 67% of insect's mortality at day 1. This emulsion caused the highest insect mortality on day 20 (Table 5) and showed significant differences compared to the other treatments ($p < 0.001$).

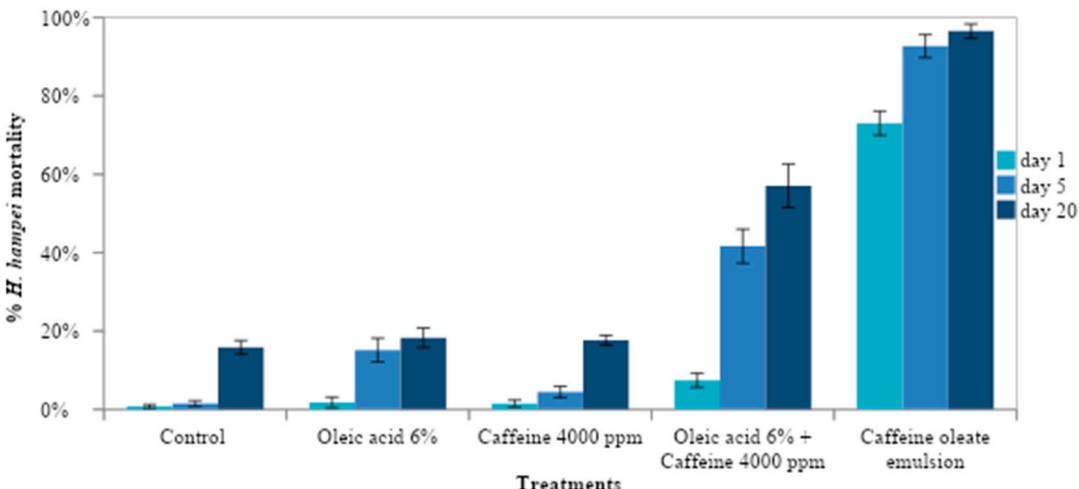

**Figure 5.** Percentage of CBB mortality evaluated on day 1, 5, and 20 after preinfestation treatment.

**Table 5.** Percentage of CBB mortality and of healthy seeds at 20 days after preinfestation treatment.

| Treatment | Experimental Unit | Mortality (%) | | | Healthy Seed (%) | | |
|---|---|---|---|---|---|---|---|
| | | Average | | Sd | Average | | Sd |
| T1. Control (Water) | 5 | 15.6 | c | 3.9 | 47.0 | c | 4.6 |
| T2. Oleic acid 6% | 5 | 18.3 | c | 5.6 | 53.0 | c | 5.1 |
| T3. Caffeine 4000 μg mL$^{-1}$ | 5 | 17.7 | c | 2.8 | 53.7 | c | 6.8 |
| T4. Oleic Acid 6% + caffeine 4000 μg mL$^{-1}$ | 5 | 57.1 | b | 2.3 | 83.7 | b | 3.2 |
| T5. Caffeine oleate emulsion (emulsion + kaolin) | 5 | 96.5 | a | 4.1 | 95.7 | a | 6.0 |

For each variable, different letters indicate differences according to Tukey 5%. Sd = Standard deviation.

Regarding the percentage of healthy seeds (Table 5), there were also differences between the control and the caffeine oleate emulsion (ANOVA, F = 84.1, DF = 4.20, $p < 0.0001$) and according to Tukey test at 5% ($p < 0.001$). In this evaluation, 47% of the seeds were found healthy on the control, while the application of the caffeine oleate emulsion was observed to protect 96% of the seeds.

The results of the postinfestation test of treatment from (Table 2) showed that none of the individual compounds had a significant effect against CBB (Figure 6). Only the caffeine oleate emulsion showed high mortality on CBB since day 1 with significant differences to the control (ANOVA, F = 44.28, DF = 4.10, $p < 0.0001$). With this emulsion, the mortality was 82% on day 20 of evaluation (Table 6).

The highest mortality caused by the caffeine oleate emulsion was reflected in the percentage of healthy seeds on day 20 of the evaluation (Table 6). The caffeine oleate emulsion protected 82% of the coffee seeds, showing differences between treatments (ANOVA, F = 10.78, DF = 4.10, $p = 0.0012$).

The effect of the caffeine oleate emulsion in the pre- and post-evaluations was higher than the caused by the mixture of caffeine and oleic acid (Tables 5 and 6). These differences were due to the elaboration process of the emulsifiable concentrate of caffeine oleate, where the formation of the caffeine oleate complex was promoted by heating the mixture of caffeine and oleic acid.

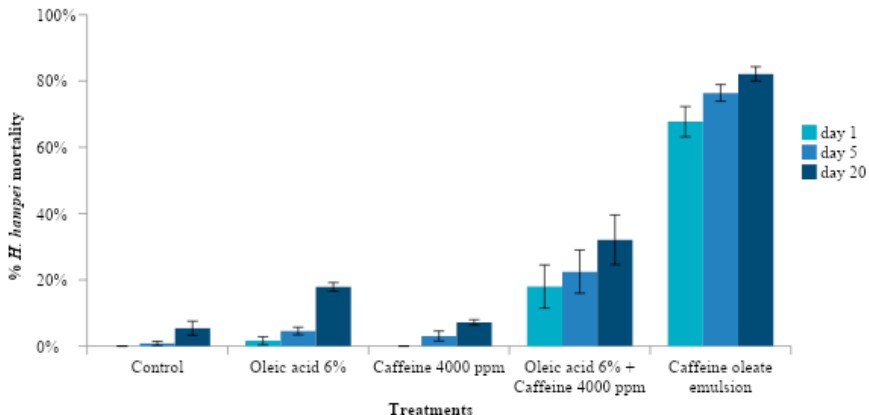

**Figure 6.** Percentage of CBB mortality evaluated on day 1, 5, and 20 after postinfestation treatment.

**Table 6.** Percentage of CBB mortality and of healthy seeds at 20 days after postinfestation treatment.

| Treatment | Experimental Unit | Mortality (%) | | | Healthy Seed (%) | | |
|---|---|---|---|---|---|---|---|
| | | Average | | Sd | Average | | Sd |
| T1. Control (Water) | 5 | 5.4 | c | 4.7 | 64.2 | b | 7.4 |
| T2. Oleic acid 6% | 5 | 17.9 | c | 2.3 | 66.7 | b | 5.2 |
| T3. Caffeine 4000 µg mL$^{-1}$ | 5 | 7.2 | bc | 1.8 | 66.3 | b | 2.2 |
| T4. Oleic Acid 6% + caffeine 4000 µg mL$^{-1}$ | 5 | 32.1 | b | 16.7 | 71.2 | b | 8.4 |
| T5. Caffeine oleate emulsion (emulsion + kaolin) | 5 | 82.2 | a | 4.8 | 91.0 | a | 3.2 |

For each variable, different letters indicate differences according to Tukey 5%. Sd = Standard deviation.

### 3.4. Evaluation of the Preventive Effect of Caffeine Oleate on CBB in the Field

For the branches sprayed with water, a large percentage of fruits were attacked by the coffee borer (Figure 7a). For the branches where the most concentrated kaolin emulsion was applied, the insects did not penetrate the fruits and avoided them in such a way that infestation did not exceed 23% after 15 days, reducing the number of fruits attacked by CBB on the trees by up to 60% compared to that in the control with water ($p < 0.0001$), ($t$ test). The toxicity of the emulsion plus kaolin to *H. hampei* was confirmed in the field, as high mortality was found (Figure 7b) that was higher than that in the control with water.

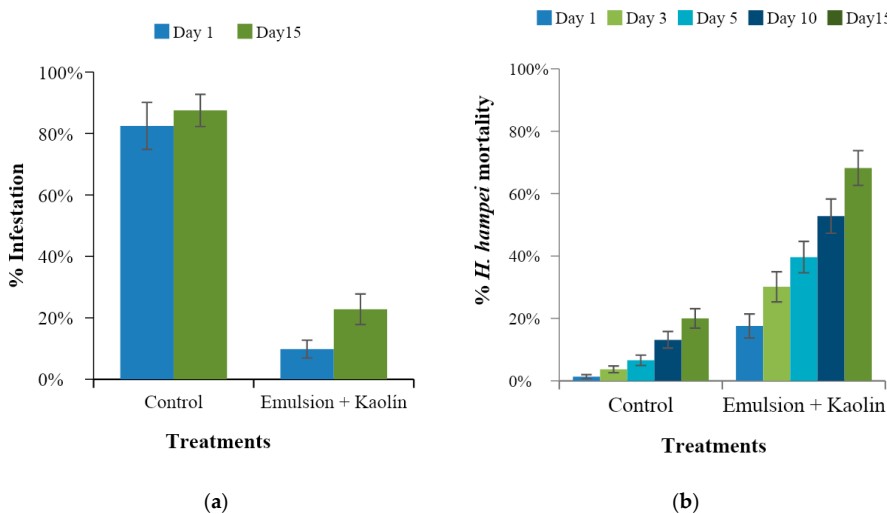

(**a**)    (**b**)

**Figure 7.** Preventive effect of caffeine oleate under field conditions. (**a**). Percentage of infestation evaluated on day 1 and day 15 after infestation. (**b**) Percentage of CBB mortality evaluated on days 1, 3, 5, 10, and 15 after infestation.

The protective effect of the emulsion was demonstrated. With respect to seed damage, for the control with water, the percentage of seeds affected by the insect was 42%, while with the emulsion, this percentage was significantly lower ($p < 0.0001$), ($t$ test), reaching only 10%. The percentage of seeds affected by CBB was 70% lower when the kaolin emulsion was applied than when the control was applied under field conditions.

### 3.5. Evaluation of the Curative Effect of Caffeine Oleate on CBB in the Field

The results of the curative effect bioassay showed 48% CBB mortality for the kaolin emulsion (Figure 8a), which was lower than the mortality induced by the same product in the preventive experiment; however, when analyzing the percentage of damaged seeds, a similar protective effect was observed. The percentage of seeds affected was only 11.5% (Figure 8b). When CBB was first entering the fruit (positions A and B), the emulsion prevented the insect from further entering the interior of the seeds, reducing damage by up to 66% compared to that in the control ($p < 0.0001$), ($t$ test).

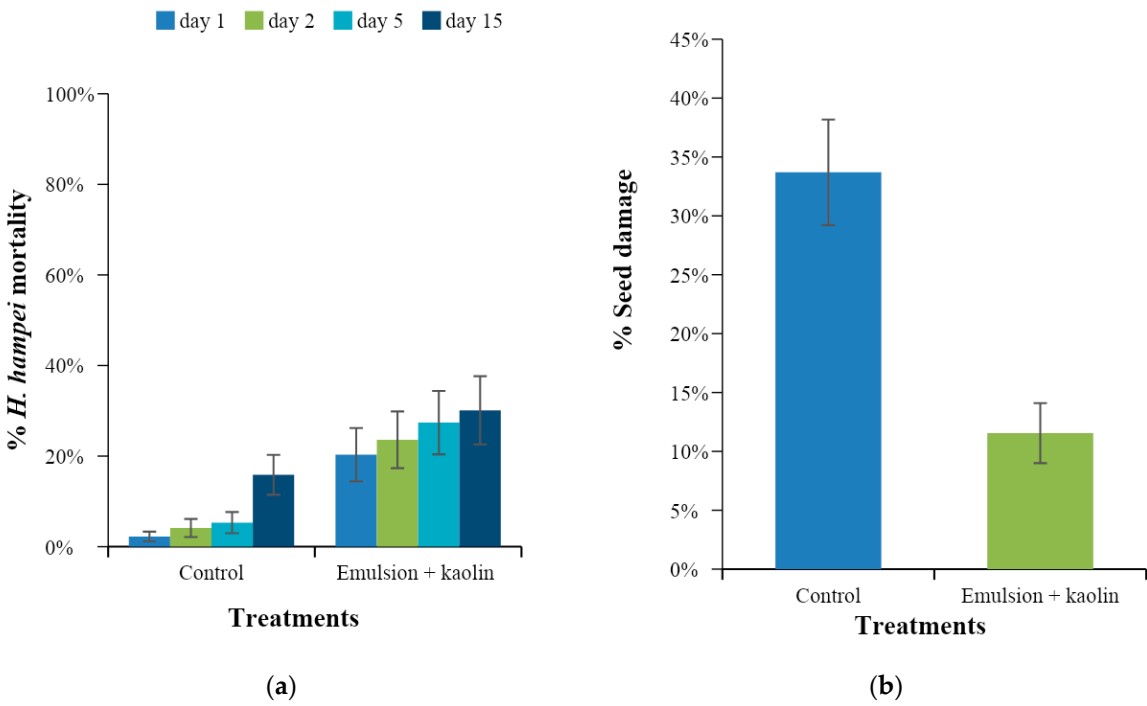

(**a**)                                                                                              (**b**)

**Figure 8.** Postinfestation effect of the emulsifiable concentrate under field conditions. (**a**) Percentage of CBB mortality evaluated on days 1, 2, 5, and 15 after infestation. (**b**) Percentage of seed damage caused by *H. hampei*.

### 3.6. Evaluation of the Curative Effect of Caffeine Oleate on M. velezangeli under Laboratory Conditions

In the evaluation of *M. velezangeli*, no natural mortality was observed in the samples sprayed with water, while spraying with the emulsion killed all the insects by day 4, with high mortality beginning on day 2 (Figure 9). Significant differences were observed (ANOVA, F = 150.71, DF = 1.16, $p < 0.0001$) The emulsion killed both nymphs and adults of the insect. All the dead insects showed signs of desiccation (Figure 10). Furthermore, no feeding marks were observed on *C. verticillate* leaves treated with caffeine oleate, in contrast to the leaves sprayed with water (Figure 10).

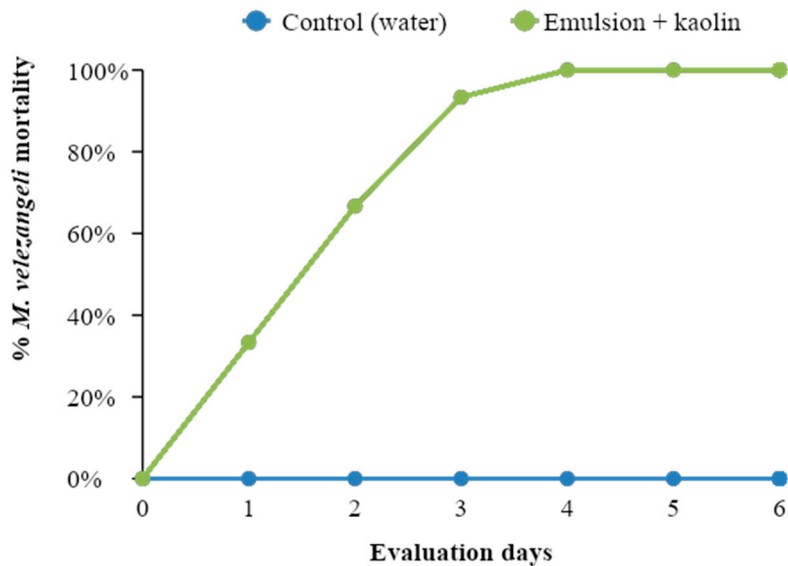

**Figure 9.** Percentage of *M. velezangeli* mortality over time.

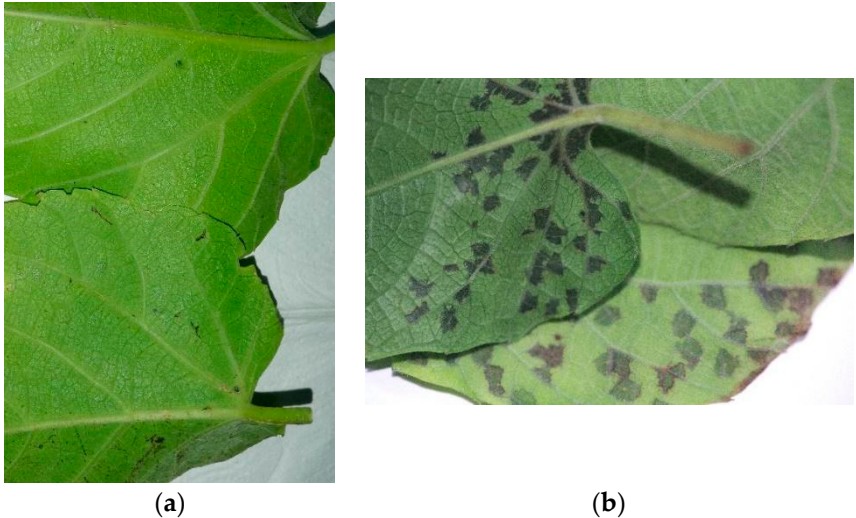

| (**a**) | (**b**) |

**Figure 10.** Signs of *C. verticillate* consumption by *M. velezangeli*. (**a**). Emulsion–kaolin treatment. No signs of feeding were observed in the control. (**b**). Control.

*3.7. Evaluation of Anatomical Changes of CBB in Contact with Caffeine Oleate by Using Scanning Electron Microscopy*

The anatomical changes induced on CBB by their contact with caffeine oleate were analyzed by scanning electron microscopy (Figure 11). A right dorso-lateral view of non-treated CBB is presented on Figure 11a showing its head, thorax, and abdomen. Figure 11d presents the ventral view of non-treated CBB, where it is possible to observe elements such as mouthparts, antenna, legs, abdominal sternites, and wings. Images of CBB sprayed with COEK are shown in Figure 11b (dorsal view) and Figure 11e (ventral view). The direct contact of CBB dorsum with COEK spray droplets induced a significant separation of CBB pronotum from the CBB elytra, increasing the distance between these two body sections up to 270 μm (Figure 11b). On the other hand, the insect ventral view shows a partial destruction of mouthparts and a significant displacement of wings moving away from the thorax (Figure 11e).

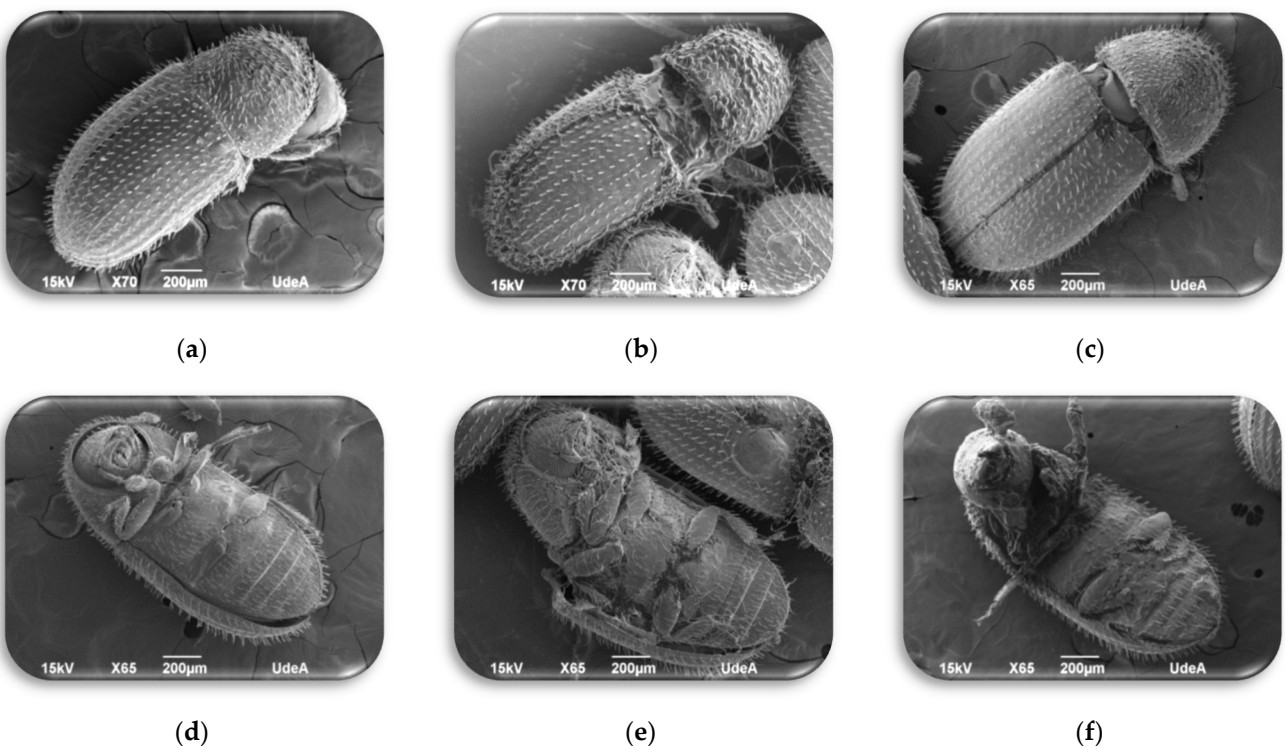

**Figure 11.** *Hypothenemus hampei* scanning electron microscopy. (**a**) Right dorso-lateral view of non-treated CBB. (**d**) Ventral view of non-treated CBB. (**b**) CBB sprayed with COEK (Caffeine oleate emulsion kaolin) dorsal view. (**e**) CBB sprayed with COEK ventral view. (**c**) CBB that were put in contact with a piece of paper wetted with COEK dorsal view. (**f**) CBB that were put in contact with a piece of paper wetted with COEK ventral view.

Images of CBB that were put in contact with a piece of paper wetted with COEK are shown in Figure 11c (dorsal view) and Figure 11f (ventral view). CBB dorsal view again indicates a thorax–abdomen separation of around 100 μm, whereas the ventral view shows the loss of most of the insect body parts (e.g., legs, mouthparts, and antennas) that were in direct contact with the COEK present on the insect walking path.

## 4. Discussion

The caffeine oleate emulsion with kaolin had a toxic effect on *H. hampei* when it was sprayed on healthy coffee fruits (preventive effect) and infested coffee fruits (curative effect) as well as *M. velezangeli* via contact under laboratory conditions. On CBB the emulsion caused high CBB mortality, preventing infestation of coffee fruits and seed damage. Under field conditions with entomological sleeves, the effects were similar to those observed in the laboratory, and caffeine oleate emulsions protected the fruits by preventing infestation and seed damage.

For both insects, the effect of the emulsion was evident 24 h after being applied. However, it took 2 to 5 days of exposure to induce mortality greater than 90%

Caffeine has not been widely used as a commercial insecticide in part due to the difficulty of its formulation. Caffeine is a hydrophilic material with a Log P of −0.07 [50]. However, it has limited solubility in water at room temperature due to the presence of three hydrophobic methyl groups and a flat structure [51]. Therefore, it does not easily penetrate lipophilic tissues, such as the cuticle of insects. The effectiveness of a caffeine-based product will thus depend on its formulation, which must allow it to penetrate the insect cuticle or overcome the barriers that it encounters in the insect's digestive system.

The use of caffeine for CBB control was initially considered unlikely, because it did not penetrate the cuticle of the insect and, if it was ingested, the insect microbiota had

the ability to degrade it in the digestive tract, conferring resistance [45]. Similarly, some hemipterans have also been shown to have the ability to detoxify the alkaloid because of the digestive enzymes that they possess [42].

Araque et al. (2007) demonstrated differences in the activity of caffeine in aqueous solutions and oil-in-water (O/W) emulsions. Caffeine in aqueous solutions has no effect on CBB via contact or ingestion. However, when formulated as an O/W emulsion, caffeine has toxic effects via both secondary contact and ingestion because it is carried in the oil phase of the emulsion, penetrating the cuticle of the insect [40].

Therefore, the effect of caffeine emulsions at a concentration of 0.4% $w/w$ caffeine and 6% $w/w$ oleic acid was evaluated, and this caffeine concentration was 10 times higher than that reported by Araque et al. (2007) [40]. Preliminary data indicated that for CBB, high survival was observed at doses below 0.1% caffeine; in *D. melanogaster*, the response depended on the delivered dose of the alkaloid [52].

On the other hand, the activity against CBB was not due to the free caffeine supplied but rather depended on the amount of oleic acid used. This occurred because the caffeine oleate molecule had the toxic effect.

The higher bioactivity of caffeine emulsions (O/W) can be explained in terms of the higher hydrophobicity of the caffeine oleate complex present in the emulsion oil phase compared to the caffeine hydrate molecule in the aqueous formulation [40,50].

The caffeine oleate complex forms due to the interaction of the hydrogen bond of carboxylic acid in oleic acid and the imidazole group of caffeine [53,54]. It is believed that this lipophilic complex is more easily absorbed through the insect cuticle [40], which generates an effect by direct contact (postinfestation) and secondary contact (preinfestation).

Additionally, oleic acid may enhance the action of caffeine on the insect cuticle, potentially because oleic acid increases the permeation of caffeine [55]. Oleic acid has been used to increase the permeation of polar and nonpolar compounds through the skin [56,57]. In addition, the oleic acid can facilitate the flow of cationic blockers through a lipoidal membrane [58].

We additionally evaluated the effect of the application of oleic acids followed by the application of caffeine, and after 15 days, we observed insect mortality of up to 30% (preliminary data). In addition, the scanning electron microscopy showed degradation of the structure between the pronotal disc and elytral disc at the elytral base (Figure 11b).

The increased activity of the emulsion with the addition of kaolin could be due to the abrasive and adsorptive properties of kaolin against epicuticular lipids [46]. Some studies have reported on the insecticidal properties and mechanisms of action of mineral particles such as kaolin [47]. It has been established that this type of material causes structural alteration or partial elimination of the epicuticle, which leads to rapid losses of water from the insect's body and mortality by desiccation [46]. The presence of kaolin in the emulsion, in addition to causing desiccation, could also allow greater penetration of the insect by the emulsion, increasing its effectiveness.

The effects of caffeine and caffeine oleate inside the insect could be related to the role of caffeine as a neurotransmitter inhibitor. Caffeine has different sites of action in vertebrate and invertebrate organisms. At the nervous system level, it blocks the adenosine receptors of subtypes A1, A2A, and A2B [59]. This blockage is responsible for its excitation effect.

Studies of *D. melanogaster* neurons have shown that caffeine inhibits the $Ca^{2+}$ current, weakly inhibits the $Na^{+}$ current, and modulates the potassium current that could increase neuronal excitability [24]. The toxicity of caffeine is also related to its negative effects on DNA repair and recombination pathways and delayed cell cycle progression, as shown in systems such as yeast and zebrafish [60,61].

Additionally, in other insects, caffeine inhibits feeding (Uefuji et al., 2005), and it paralyzes and has toxic effects on insects by inhibiting phosphodiesterase activity and by increasing intracellular levels of cyclic adenosine monophosphate (AMP) [62,63]. At the muscle cell level, caffeine affects the ryanodine receptor channels that regulate calcium release and are essential for feeding, flight, locomotion, and oviposition [64]. Binding

of caffeine to the receptor increases the affinity of the $Ca^{2+}$ receptor, leading to channel activation at low $Ca^{2+}$ levels [65].

The formulation based on caffeine and kaolin oleate, highly effective in controlling CBB under laboratory and field conditions, could also be used in the control of other agricultural pest insects. This prototype is scalable, economically viable, and potentially environmentally friendly. In addition, since the components of the formulation are food grade and are generally recognized as safe (GRAS) by the US FDA, the product would be safer than other options for both the coffee ecosystem and the health of coffee grower.

## 5. Conclusions

A new caffeine oleate formulation was developed that showed laboratory efficacy by causing mortality of more than 90% of CBB adults in preventive tests in which the insecticide was sprayed prior to insect attack on the coffee fruits. In the curative tests, in which spraying occurred after CBB infested the fruits, the formulation caused 77% mortality of the insects. Under controlled field conditions with artificial infestations, the product kept CBB infestation below 20%, reducing the number of fruits attacked by the insect by up to 70% compared to the numbers in the controls. In addition, no phytotoxic effects were observed in coffee plants. With respect to *M. velezangeli*, the insecticide was effective, causing 100% mortality. Similarly, the emulsion of caffeine oleate with kaolin had toxic effects and a marked effect on *M. velezangeli* via contact. For both insects, the effect of the emulsion was evident 24 h after being applied.

The results revealed a correlation between the caffeine-oleic acid relationship and toxicity to the insect, showing that the activity against CBB was not due to the free caffeine supplied but rather depended on the amount of oleic acid used. This occurred because the caffeine oleate molecule had the toxic effect.

This phytochemical formulation could be included as part of a strategy for integrated management of CBB as well as other pests of coffee.

## 6. Patent

Benavides Machado, P., Casanova, Y., Góngora, C.E., González, S., and Tapias, L.J. (2022). Concentrados emulsificables que comprenden un sistema sobresaturado de cafeína, ácidos grasos y surfactantes que presentan actividad insecticida (Superintendencia de Industria y Comercio Patente Núm. NC2019/0010015).

**Author Contributions:** Conceptualization, methodology, and supervising the work, H.C., P.B. and C.E.G.; writing the original draft, C.E.G.; investigation and writing, J.T. and J.J.; formal statistical analysis, R.M.; methodology and investigation, T.R.; resources, S.G. and H.C.; funding acquisition, H.C. and P.B. All authors have read and agreed to the published version of the manuscript.

**Funding:** This research was funded by Colciencias, the company Nexentia-Sumicol S.A, the National Federation of Coffee Growers of Colombia, and Universidad de Antioquia, under the project numbers ENT102003 and ENT107012 and the contracts RC-0162-2014 and CN-2019-0310.

**Data Availability Statement:** Not applicable.

**Acknowledgments:** The authors thank the insect breeding unit of Biocafé for providing coffee berry borers. The authors are also grateful to the Agronomy Engineer Myriam Cañon of the Experimental Station Paraguaicito for assisting with field evaluations.

**Conflicts of Interest:** The authors declare no conflict of interest.

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
