# Peer review of "A Novel Caffeine Oleate Formulation as an Insecticide to Control Coffee Berry Borer, Hypothenemus hampei, and Other Coffee Pests"

_agronomy, doi:10.3390/agronomy13061554_

Round 1
Reviewer 1 Report
In the current study, the authors mainly explored the effects of a novel caffeine oleate formulation on the control of the coffee berry borer (CBB), Hypothenemus hampei (Ferrari) and the tropical plant bug, Monalonion velezangeli (Carvalho and Costa) in order to find out an alternative for the chemical control based on the commonly-used insecticides. Their findings will greatly contribute to provide valuable information for the biocontrol of insect pests on coffee. However, the methods used in the manuscript and the results should be concise and accurate. Additionally, this manuscript has some errors listed in the major and minor concerns which need to be addressed as follow:
Major concerns:
1. To improve the systematicity, completeness and conciseness of the present study and make it easily understand for readers, I suggest the authors should delete the contents related to the curative effect of caffeine oleate on M. velezangeli. Of course, its corresponding results could be discussed about the curative effect of caffeine oleate on H. hampei.
2. In the sections of Materials and Methods, I strongly recommend the authors should provide the information about the volume of spray application for “2.2” and “2.8” (Line 286) and of COEK for wetting a piece of paper. In addition, they should simplify the same procedures (investigation date and method) of different experiments. For example, Lines 246-250 could be simplified.
3. For “2.4.”of Materials and Methods, the authors should describe the investigation date. Additionally, most of the investigation dates in the Materials and Methods are not consistence with the corresponding results in Figures 3-9. Please check them carefully.
4. For the method of statistical analysis, I don’t know why the authors adopt three different tests including Dunnett’s 5% comparison test (Line 185), Turkey’s 5% comparison test (Line 202) and Duncan’s test (Line 231) to compare significant differences among different treatments.
5. For Tables3-6 and Figures 3-9, their titles should be concise and accurate. For example, the title of Table 3 could be revised as “Percentage of CBB mortality and of healthy seeds at 20 days after preinfestation treatment”.
6. Throughout the manuscript, the authors should replace “ppm” and “ug mL-1” with “μg mL-1” as well as for Tables 1-2 and 5-6. For Tables 1-3 and 5-6, “,” should be corrected as “.”. Additionally, the full name of “standard deviation” should be provided for the abbreviation of “Sd” in the notes of Tables 3-6.
7. For insect nomenclature, I suggest the authors should follow “International Code of Zoological Nomenclature (Fourth Edition)”.
8. For Discussion and Conclusions, some paragraphs should be integrated. For example, first and second paragraphs (Lines 478-487), fourth to sixth paragraphs (Lines 495-511), seventh and eighth paragraphs (Lines 512-518), ninth to eleventh paragraphs (Lines 519-531), thirteenth to fifteenth paragraphs (Lines 540-556), thirteenth to fifteenth paragraphs (Lines 540-556) of Discussion, and first to second paragraphs (Lines 564-576) and third to fourth paragraphs (Lines 578-585) could be integrated into new paragraph, respectively.
9. For References, I strongly recommended the authors should re-write according to MDPI Reference List and Citations Style Guide (https://www.mdpi.com/authors/references). Especially, these words in the reference titles don’t need to be capitalized the first letter except for the first word or special noun.
Minor concerns:
1. In Lines 16 and 34, “The coffee berry borer” is more better than “Coffee berry borer”.
2. In Lines 59, 545-46 and 555-556, “Ca+2” and “Na+” should be revised as “Ca+2”, and “Na+”, respectively.
3. In Line 76, delete the Latin name of tea because of other plants without their Latin names.
4. In Lines 80-81, “X. fornicates” should be corrected as “X. fornicatus”, respectively.
5. In Line 90, “;” should be corrected as “,”.
6. In Lines 91, 387 and 487, “.” should be added at the end of a sentence”.
7. In Line 108, “TL50” should be revised as “TL50”.
8. In Line 130, “801180” should be wrong. Please check it.
9. In Lines 220-221, delete the second “, Soacha, Cundinamarca, Colombia” because it appeared again.
10. In Lines 281 and 450, “coffee berry borer” should be abbreviated as “CBB”.
11. In Line 286, “the caffeine oleate emulsion with added kaolin” should be revised as “the caffeine oleate emulsion including kaolin (COEK)”. In Lines 288 and 457, both “the caffeine oleate emulsion with added kaolin (COEK)” and “COEK (caffeine oleate emulsion kaolin)” should be revised as “COEK”.
12. In Line 300, “Table 2” should be corrected as “Table 3”.
13. In Line 346, “(P<0.001).)” should be revised as “(P<0.001)”.
14. In Line 365, “show” should be corrected as “showed”.
15. In Lines 389, 420, 434-435 and 451-452, don’t be in bold for these subtitles.
16. In Lines 478, 485 and 510, these Latin names of insect species should be in italic.
17. In Line 499, “42” should be corrected as “[42].”. Additionally, “Araque 40” should be corrected as “Araque et al. (2007) [40].”.
18. In Line 531, “Figure 11b” should be corrected as “(Figure 11b)”.
19. In Lines 584-585, delete the second sentence.
20. For “6. Patent”, it is better as a reference.
Finally, I hope the authors can use these to correct the same problem for the rest.
The quality of English language in the manuscript is moderate and I recommend the authors should edit carefully.
Author Response
Agronomy-2411699
The authors would like to thank the reviewer for all the comments that helped us to improve the quality and clarity of the manuscript. In the new version of the paper all the addition are identified in red color and all the deletion were left identified in order to the review and editor to see them.
Each one of the comments were answering
Reviewer1
Major concerns:
- To improve the systematicity, completeness and conciseness of the present study and make it easily understand for readers, I suggest the authors should delete the contents related to the curative effect of caffeine oleate on velezangeli. Of course, its corresponding results could be discussed about the curative effect of caffeine oleate on H. hampei
Authors reply
- We will like to wait for the opinion from the other reviewers because we think that information related to velezangeli is important in order to understand the manuscripts as a whole.
- Reviewer 2 make some suggestion about improve Figure 9 and 10. It seems that that reviewer as well as us, consider important to leave this information in the manuscript
2 . In the sections of Materials and Methods, I strongly recommend the authors should provide the information about the volume of spray application for “2.2” and “2.8” (Line 286) and of COEK for wetting a piece of paper. In addition, they should simplify the same procedures (investigation date and method) of different experiments. For example, Lines 246-250 could be simplified.
Authors reply.
- When the spray applications were carried out with a portable sprayer in all the cases the volume application was of 500 u The information was added in line 147-148 and line 172
- Respect to the information to COEK. – the insects were sprayed with 300 uL of the caffeine oleate emulsion with added kaolin and the last group insects that were in contact with the insecticide by walking on a piece of towel paper, that cover the base of a petri dish, wetted with 400 uL of (COEK). The information was added in lines 290 to 293
- Lines 251-254 were deleted and instead, this information was added: -analyses was done as describe previously for 2.5.
- For “2.4.”of Materials and Methods, the authors should describe the investigation date. Additionally, most of the investigation dates in the Materials and Methods are not consistence with the corresponding results in Figures 3-9. Please check them carefully.
Authors reply
- We check all the data and Figures. We made de corrections and clarify the information
- Figure 3 corresponds to treatment describe in table 1- We correct the information in line 304 change Table 2 to Table 3
- Line 330 we add information (table 1)
- Line 357 we add information (treatment from table 2)
- Line 382 we add information (treatment from table 2)
- For the method of statistical analysis, I don’t know why the authors adopt three different tests including Dunnett’s 5% comparison test (Line 185), Turkey’s 5% comparison test (Line 202) and Duncan’s test (Line 231) to compare significant differences among different treatments.
Authors reply
- The statistical analysis was different depending on the number of treatments and type of comparison.
- Dunnett's 5% comparison test was performed to compare the 2 treatments vs. absolute control (water control) table 1. This test was performed for 2.3 Curative effect of caffeine oleate on CBB under laboratory conditions. Here we only were interest in comparing each one of the treatments vs control.
- Since the treatments were significantly different from the control, a 5% least significant difference (LSD) test was applied to compare the two emulsions. Information in Line185-188.
- Tukey comparison test was performed to compare 5 treatments (table 2). All the treatments are compared among each other. 4 Effect of caffeine oleate emulsion components against CBB under laboratory conditions. Our interest was to identify the best one of the treatments.
- Respect to Duncan test the reviewer is right. We only performed a descriptive analysis of the data as is showed in Figure 7 and 8. The effect on the insects were quite evident. Because of this the information about Duncan test was deleted in line 234-235
- We add the information about a T test to explain the probability (P <0.0001)- Line 416-Line 426 and Line 437.
- For Tables3-6 and Figures 3-9, their titles should be concise and accurate. For example, the title of Table 3 could be revised as “Percentage of CBB mortality and of healthy seeds at 20 days after preinfestation treatment”.
Authors reply
- The titles from Tables and Figure suggested by the reviewers were changed in
Table 3- Table 4- Table 5 -Table 6-
- Fig 3- Fig 4 - Fig 5- Fig 6- Fig 7- Fig 8 -Fig 9.-
- Throughout the manuscript, the authors should replace “ppm” and “ug mL-1” with “μg mL-1” as well as for Tables 1-2 and 5-6. For Tables 1-3 and 5-6, “,” should be corrected as “.”. Additionally, the full name of “standard deviation” should be provided for the abbreviation of “Sd” in the notes of Tables 3-6.
Authors reply
- We change ppm to μg mL-1 Line 207 . Table 2. Table 5- Table 6
- We change “ug mL-1 to μg mL-1 Table 1 and Table 5
- Sd = Standard deviation. We add the information in Table 3- Table 4. Table5 and Table 6
- For insect nomenclature, I suggest the authors should follow “International Code of Zoological Nomenclature (Fourth Edition)”.
Following the recommended nomenclature the name of the insects is as follow:
Authors reply
- Hypothenemus hampei(Ferrari, 1867) (Coleoptera:Curculionidae:Scolytinae).
- Monalonion velezangeliCarvalho & Costa, 1988 (Hemiptera:Miridae).
- When a species has been transferred or changed to a different genus since its original description, the author's name of the species is cited in parentheses. In the case of the coffee berry borer, it was originally described in the genus Cryphalus in 1867 and later transferred to the genus Stephanoderes in 1871.
- In the case of Monalonion, it has not been changed, so the author is quoted without parentheses.
- We made the correction in Line 16 and 35 for CBB and for Monalonion in line 21 and 45
- For Discussion and Conclusions, some paragraphs should be integrated. For example, first and second paragraphs (Lines 478-487), fourth to sixth paragraphs (Lines 495-511), seventh and eighth paragraphs (Lines 512-518), ninth to eleventh paragraphs (Lines 519-531), thirteenth to fifteenth paragraphs (Lines 540-556), thirteenth to fifteenth paragraphs (Lines 540-556) of Discussion, and first to second paragraphs (Lines 564-576) and third to fourth paragraphs (Lines 578-585) could be integrated into new paragraph, respectively.
Authors reply
- (Lines 478-487) were integrated
- (Lines 495-511 and 512-518 part of the information was delete to have more integration.
- Lines 519-531- Line 540 to 556- This information was not change because it explains the mode of action of the insecticide and it is very important to undertint the manuscript.
- First to second paragraphs (Lines 564-576) and third to fourth paragraphs (Lines 578-585) could be integrated into new paragraph, respectively. It was done
9.For References, I strongly recommended the authors should re-write according to MDPI Reference List and Citations Style Guide (https://www.mdpi.com/authors/references). Especially, these words in the reference titles don’t need to be capitalized the first letter except for the first word or special noun.
Authors reply
- We check the example
Journal references must cite
- Bowman, C.M.; Landee, F.A.; Reslock, M.A. Chemically Oriented Storage and Retrieval System. 1. Storage and Verification of Structural Information. J. Chem. Doc. 1967, 7, 43-47; DOI:10.1021/c160024a013.
The words in the reference titles are capitalized. However, each one of the references was check again and the changes when it were necessary were done.
Minor concerns:
- In Lines 16 and 34, “The coffee berry borer” is more better than “Coffee berry borer”.
- Change was done
- In Lines 59, 545-46 and 555-556, “Ca+2” and “Na+” should be revised as “Ca+2”, and “Na+”, respectively.
- Change was done
- In Line 76, delete the Latin name of tea because of other plants without their Latin names.
All the plants were identified with the Latin name
- In Lines 80-81, “X. fornicates” should be corrected as “ fornicatus”, respectively.
- Change was done
- In Line 90, “;” should be corrected as “,”.
- Change was done
- In Lines 91, 387 and 487, “.” should be added at the end of a sentence”.
- Change was done
- In Line 108, “TL50” should be revised as “TL50”.
- Change was done
- In Line 130, “801180” should be wrong. Please check it.
- It is 80°C the change was done
- In Lines 220-221, delete the second “, Soacha, Cundinamarca, Colombia” because it appeared again.
- Change was done
- In Lines 281 and 450, “coffee berry borer” should be abbreviated as “CBB”.
- Change was done
- In Line 286, “the caffeine oleate emulsion with added kaolin” should be revised as “the caffeine oleate emulsion including kaolin (COEK)”. In Lines 288 and 457, both “the caffeine oleate emulsion with added kaolin (COEK)” and “COEK (caffeine oleate emulsion kaolin)” should be revised as “COEK”.
- Change was done
- In Line 300, “Table 2” should be corrected as “Table 3”.
- Change was done
- In Line 346, “(P<0.001).)” should be revised as “(P<0.001)”.
- Change was done (Line 359)
- In Line 365, “show” should be corrected as “showed”.
- Change was done (line 382)
- In Lines 389, 420, 434-435 and 451-452, don’t be in bold for these subtitles.
- The subtitles 3.4, 3.5, 3.6 and 3.7 were corrected
- In Lines 478, 485 and 510, these Latin names of insect speciesshould be in italic.
- Change was done
- In Line 499, “42” should be corrected as “[42].”. Additionally, “Araque 40” should be corrected as “Araque et al. (2007) [40].”.
- Change was done
- In Line 531, “Figure 11b” should be corrected as “(Figure 11b)”.
- Change was done
- In Lines 584-585, delete the second sentence.
- Change was done (Line592-593)
- For “6. Patent”, it is better as a reference.
- There is not reference there is a patent

Reviewer 2 Report
The work of Góngora end colleagues entitled "A novel caffeine oleate formulation as an insecticide to control coffee berry borer, Hypothenemus hampei, and other coffee pests" aimed to evaluate the efficacy of a caffeine-based product against Hypothenemus hampei and, partially, Monalonion velezangeli. In general, the work is focused on a very interesting topic and, considering the amount of data the authors reported, should be considered for publication in a scientific journal as Agronomy. However, I believe that the authors must improve some part of the manuscript, especially statistical analysis. For this reason, aware that authors can easily correct the manuscript, I suggest to publish the manuscript after a major revision. Attached you cand find the manuscript with numerous comments and suggestions.

No specific comment on the quality of English
Author Response
Agronomy-2411699
The authors would like to thank the reviewer for all the comments that helped us to improve the quality and clarity of the manuscript. In the new version of the paper all the addition are identified in red color and all the deletion were left identified in order to the review and editor to see them.
Each one of the comments were answered.
Specially, respect to reviewer 2 the information that was specifically ask for, it was added to the introduction and it is mark with a yellow color.
Reviewer 2
Suggestions:
Line 135: This is the first time authors mention the use of kaolin. The authors need to state in the introduction the value of kaolin as an insecticide or repellent or use as an additive to caffeine oleate and why it might increase the efficacy of caffeine oleate. The authors state some their reasoning in the discussion lines 532-9 as follows:
“The increased activity of the emulsion with the addition of kaolin could be due to the abrasive and adsorptive properties of kaolin against epicuticular lipids. [57] Some studies have reported on the insecticidal properties and mechanisms of action of mineral particles such as kaolin [58]. It has been established that this type of material causes structural alteration or partial elimination of the epicuticle, which leads to rapid losses of water from the insect's body and mortality by desiccation [57]. The presence of kaolin in the emulsion, in addition to causing desiccation, could also allow greater penetration of the insect by the emulsion, increasing its effectiveness.
Authors reply
- The information related to the Kaolin insecticidal properties and the explanation od why it was used as part of the insecticide was added to the introduction, the change and addition of information was done in line 114-120.
- Due to this all the references from the 45 to 58 were changed too.
Line 352 change word: …47% of the seeds were found healthy
Authors reply
- The change was done (Line 367)
Line 353: change word: … was observed to protect 96%
Authors reply
- The change was done (Line 368)
Line 439-441: rewrite to cite correct Figures
…signs of desiccation (Figure 9). Furthermore, no feeding marks were observed on C. verticillata leaves treated with caffeine oleate, in contrast to the leaves sprayed with water (Figure 10)
Authors reply
- The change was done Figure 8 was change to Figure 10 (Line 451)

Reviewer 3 Report
Wednesday, May 26, 2023
Manuscript ID: agronomy-2411699
Type Article : Title A novel caffeine oleate formulation as an insecticide to control coffee berry borer, Hypothenemus hampei, and other coffee pests
Authors: Carmenza E. Góngora * , Johanna Tapias , Jorge Jaramillo , Rubén Medina , Sebastián González , Tatiana Restrepo , Herley Casanova , Pablo Benavides
Section: Pest and Disease Management
Special Issue Sustainable Strategies for the Control of Crop Diseases and Pests to Reduce Pesticides
Abstract Coffee berry borer (CBB), Hypothenemus hampei (Ferrari) (Coleoptera: Curculionidae: Scolytinae), is the pest that causes the most economic damage to coffee crops. Chemical control of this insect is based on the use of insecticides that can affect the environment and nontarget organisms. Despite the fact that caffeine has shown potential as an insecticide, a caffeine-based product for field use is currently not available on the market. As a new alternative to control CBB and other coffee pests, such as Monalonion velezangeli, a caffeine-oleate was developed. The caffeine oleate formulation showed laboratory efficacy by causing mortality of more than 90% of CBB adults in preventive tests in which the insecticide was sprayed prior to insect attack on the coffee fruits. In the curative tests, in which spraying occurred after CBB infested the fruits, the formulation caused 77% mortality of the insects. Under controlled field conditions, the product kept CBB infestation below 20%, reducing the number of fruits attacked by the insect by up to 70%. In addition, no phytotoxic effects were observed in coffee plants. The insecticide was also effective against M. velezangeli causing 100% mortality. A caffeine oleate formulation that could be part of a strategy for integrated CBB management as well as other pests of coffee was developed. The components of the insecticide are food grade; the product would provide greater security to the coffee ecosystem and coffee growers.
Suggestions:
Line 135: This is the first time authors mention the use of kaolin. The authors need to state in the introduction the value of kaolin as an insecticide or repellent or use as an additive to caffeine oleate and why it might increase the efficacy of caffeine oleate. The authors state some their reasoning in the discussion lines 532-9 as follows:
“The increased activity of the emulsion with the addition of kaolin could be due to the abrasive and adsorptive properties of kaolin against epicuticular lipids. [57] Some studies have reported on the insecticidal properties and mechanisms of action of mineral particles such as kaolin [58]. It has been established that this type of material causes structural alteration or partial elimination of the epicuticle, which leads to rapid losses of water from the insect's body and mortality by desiccation [57]. The presence of kaolin in the emulsion, in addition to causing desiccation, could also allow greater penetration of the insect by the emulsion, increasing its effectiveness.
Line 352 change word: …47% of the seeds were found healthy
Line 353: change word: … was observed to protect 96%
Line 439-441: rewrite to cite correct Figures
…signs of desiccation (Figure 9). Furthermore, no feeding marks were observed on C. verticillata leaves treated with caffeine oleate, in contrast to the leaves sprayed with water (Figure 10).
Author Response
Agronomy-2411699
The authors would like to thank the reviewer for all the comments that helped us to improve the quality and clarity of the manuscript. In the new version of the paper all the addition are identified in red color and all the deletion were left identified in order to the review and editor to see them.
Each one of the comments were answered.
Specially, respect to reviewer 2 the information that was specifically ask for, it was added to the introduction and it is mark with a yellow color.
Reviewer
Suggestions:
Line 135: This is the first time authors mention the use of kaolin. The authors need to state in the introduction the value of kaolin as an insecticide or repellent or use as an additive to caffeine oleate and why it might increase the efficacy of caffeine oleate. The authors state some their reasoning in the discussion lines 532-9 as follows:
“The increased activity of the emulsion with the addition of kaolin could be due to the abrasive and adsorptive properties of kaolin against epicuticular lipids. [57] Some studies have reported on the insecticidal properties and mechanisms of action of mineral particles such as kaolin [58]. It has been established that this type of material causes structural alteration or partial elimination of the epicuticle, which leads to rapid losses of water from the insect's body and mortality by desiccation [57]. The presence of kaolin in the emulsion, in addition to causing desiccation, could also allow greater penetration of the insect by the emulsion, increasing its effectiveness.
Authors reply
- The information related to the Kaolin insecticidal properties and the explanation od why it was used as part of the insecticide was added to the introduction, the change and addition of information was done in line 114-120.
- Due to this all the references from the 45 to 58 were changed too.
Line 352 change word: …47% of the seeds were found healthy
Authors reply
- The change was done (Line 367)
Line 353: change word: … was observed to protect 96%
Authors reply
- The change was done (Line 368)
Line 439-441: rewrite to cite correct Figures
…signs of desiccation (Figure 9). Furthermore, no feeding marks were observed on C. verticillata leaves treated with caffeine oleate, in contrast to the leaves sprayed with water (Figure 10)
Authors reply
- The change was done Figure 8 was change to Figure 10 (Line 451)
Round 2
Reviewer 1 Report
The authors have revised the manuscript according to my suggestions and answered these questions point by point. I’m satisfied with almost all revisions they have made.
However, an error of punctuation marks should be corrected in Tables 1-3 and 5-6, i.e., the comma “,” should be replaced with full point “.”. Additionally, the first letter of each word in the reference titles don’t need to be capitalized except for the first word or special noun. The Latin name of a specie should be in italic.
So I think this manuscript needs to be addressed with minor revision.
Author Response
The authors would like to thank the reviewer for all the comments that helped us to improve the quality and clarity of the manuscript. In this second round of revision we make the changes accepted for the reviewer in the tables. The accepted changes are indicated red.
Each one of the comments of this second round were answering
Reviewer 3
- The authors have revised the manuscript according to my suggestions and answered these questions point by point. I’m satisfied with almost all revisions they have made.
Author Reply
- It is necessary to specify where "producers use insectides without prior evaluation- Line 49
Author Reply
- On the document I accepted all the deletion were left identified in order to the review and editor to see them in order to have a final document more organized
However, an error of punctuation marks should be corrected in Tables 1-3 and 5-6, i.e., the comma “,” should be replaced with full point “.”. Additionally, the first letter of each word in the reference titles don’t need to be capitalized except for the first word or special noun. The Latin name of a specie should be in italic.
So I think this manuscript needs to be addressed with minor revision.
- However, an error of punctuation marks should be corrected in Tables 1-3 and 5-6, i.e., the comma “,” should be replaced with full point “.”.
Authors reply
- In table 1 0,1 was change to 0.1 the comma “,” was replaced with full point
- A point (.) was added to the end of line150
- In table 2 0,1 and 0,3 was change to 0.1 the comma “,” was replaced with full point
- A point (.) was added to the end of line207
- In table 3, 5 and 6 In all the number the coma “,” was change to point.
- Additionally, the first letter of each word in the reference titles don’t need to be capitalized except for the first word or special noun. The Latin name of a specie should be in italic.
Authors reply
- We check the example
Journal references must cite
- Bowman, C.M.; Landee, F.A.; Reslock, M.A. Chemically Oriented Storage and Retrieval System. 1. Storage and Verification of Structural Information. J. Chem. Doc. 1967, 7, 43-47; DOI:10.1021/c160024a013.
The words in the reference titles are capitalized. However, each one of the references was check again and the changes when it were necessary were done.

Reviewer 2 Report
After re-evaluating the work of Gongora and colleagues I came to the conclusion that my previous report has not been received yet as no responses to my comments/suggestions are present in the file attached by the authors. I'm really sorry but I'm forced to suggest the publication of the work after a major revision. As I mentioned in the previous draft, although the manuscript took into account a very interesting topic, several parts must be corrected.
Comments and suggestions are present in the attached file

No specific comment
Author Response
The authors would like to thank the reviewer 3, for all the comments that helped us to improve the quality and clarity of the manuscript. We would like to apology to the reviewer because in the first round of review, the comments of this reviewer where not in the system. We could not see them.
Yesterday with the second version of the manuscript we could see the comments of the review3.
In the manuscript presented some of the changes that reviewer 3 suggest are indicated in green, and other additions or deletion are indicated in red underline in order to be accepted if the reviewer agree.
Each one of the comments were answering
Reviewer3
- Line 17 and 40. These products are classified into Category III following what? Please add more information
Authors reply
- The category III corresponds to the World Health Organization (WHO) classification. The information was added in line 40-41,
- Line 50. It is necessary to specify where "producers use insectides without prior evaluation..."
Authors reply
- The information was clarified, a change and addition of information was added to the sentence. The information added-and some producers from some Colombian avocado-coffee zone use insecticides without prior evaluation of their effects
- Line 69-70. Caffeine can be considered as a safe product under certain limits. Please add more information
Authors reply
- The information relate to the safety of caffeine is in line 71 to 73-….. and in the discussion the topic is explain again line 566-569
- Line 77-78 This sentence sounds a little bit strange. Caffeine should have an effect towards "feeding and reproduction" of insect rather than plants
Authors reply
- Review is right there is a mistake and the sentence was changed.
From Lycopersicon esculentum and Manduca sexta To caffeine and other methylxanthines interfere with the feeding and reproduction of Manduca sexta on Lycopersicon esculentum
- Please report the scientific name in italic- Line 81 and 83
Authors reply.
The change was done X. fornicates
- Line 117-118 This sentence sounds as incomplete
Authors reply
- The original sentences was change From “its efficacy in the control of CBB and other coffee pests was determined and diverging hyphothesis were necessary”
To: its efficacy in the control of CBB and other coffee pests was determined.
- Line 133- I believe that this temperature is too high. Please correct
Authors reply
- It is 80°C the change was done
- Line 149
Would have expected the authors to have also tested the emulsifier alone to assess any effects against CBB. As reported, unfortunately, I don't know whether the insecticidal effect is due to the caffeine or the other components of the emulsion.
Authors reply
- We did that in the following experiment, in the table 2 the treatments are showed. We test the different component of the emulsion and the results are showed in Figure 5, together with the explanation.
- Line 163
What does curative mean? I suggest to correct with either "toxity" or "direct toxicity" on CBB
Authors reply
In previous work, wich we included the references, we have used the term curative effect to refer to the effect of a insecticide o biological controller on the CBB when it is apply on the CBB, when the insect is penetrating the coffee berry and it is in position A-B.We will like to continue using the same terminology.
References
- Tapias I., L.J.; Martinez D., C.P.; Benavides M., P.; Gongora B., C.E. Método de laboratorio para evaluar el efecto de insecticidas sobre la broca del café. Laboratory method to evaluate the effect of insecticides on coffee berry borer. Cenicafé 2017. 68, 76-89.
- Góngora, C. E.; Tapias, J.; Martínez, C. P.; Benavides, P. Methodology to Test Control Agents and Insecticides Against the Coffee Berry Borer Hypothenemus hampei. J. Vis. Exp: Jove 2022, 181, e63694,doi:10.3791/63694.
- I did not understand the meaning of "position A" and "position B". More information is needed
Authors reply
CBB in Position A and B is what is showed in Figure1. The insect is penetrating the coffee berry but still a part of the abdomen is exposed, so the insecticide can reach it. In the references 48 and 49 it is showed what is position A and B. We change the reference in line 168 Benavides for [48 49]
- Line 185 - 186
Delete -The statistical analysis consisted of estimating the average and standard deviation of each response variable for each treatment.
Authors reply
- We delete and reorganized the sentence:
The average and standard deviation of each response variable for each treatmen was estimated.
- Line 184 to 190
Here and hereafter: ANOVA can be considered as corrected only if data follow assumptions of normality. I realized that authors analyzed percentage without transformation. The latter could be considered uncorrected. Please, test normality and apply transformation before analysis if required. Optionally, authors can use other statistical models such as Kruskal-Wallis test or GLM with binomial error distribution.
Please use Tukey's test for mean separation (it's more rigorous than LSD)
Authors reply
We test data normality and in all the cases the data follow a normal distribution.
It was not necessary to do transformation because the assumption of normality and homogeneity of the residuals were met
Additionally, the Kruskal-Wallis test is giving us the same information tan the ANOVA we believe that is not necessary because we dont have rank data.
In general for the statistical analysis data were expressed as means SD (standard deviation) for each one of the variables. Effect of treatments was analyzed using ANOVA according to the model for a complete randomized design at 5% and statistical significance of P<0.05
Additionally,
- Dunnett's 5% comparison test was performed to compare the 2 treatments vs. absolute control (water control) table 1. This test was performed for 2.3 Curative effect of caffeine oleate on CBB under laboratory conditions. Here we only were interest in comparing each one of the treatments vs control.
- Since the treatments were significantly different from the control, a 5% least significant difference (LSD) test was applied to compare the two emulsions. Information in Line185-188.
- Tukey comparison test was performed to compare 5 treatments (table 2). All the treatments are compared among each other. 4 Effect of caffeine oleate emulsion components against CBB under laboratory conditions. Our interest was to identify the best one of the treatments.
- Respect to Duncan test (Line 231. Experiments 2.5 and 2.6 ) other reviewer told us that was not correct. Because we only performed a descriptive analysis of the data as is showed in Figure 7 and 8. The effect on the insects were quite evident. Due to this the information about Duncan test was deleted.
- We add the information about a T test to explain the probability (P <0.0001)- Line 416-Line 426 and Line 437.
- Line 185 to 191 Delete -The statistical analysis consisted of estimating the average and standard deviation of each response variable for each treatment
Authors reply
We accepted the deletion and we add the following information:
Data were expressed as means and SD (standard deviation) for each one of the variables. Effect of treatments was analyzed using ANOVA according to the model for a complete randomized design at 5% and statistical significance of P<0.05
- Line 206-2011 Delete -The statistical analysis consisted of estimating the average and standard deviation of each response variable for each treatment.
Authors reply
We accepted the deletion and we add the following information:
Data were expressed as means and SD (standard deviation) for each one of the variables. Effect of treatments was analyzed using ANOVA according to the model for a complete randomized design at 5% and statistical significance of P<0.05
- Line 265 Thirty (30) insects divided by 5 insect is equal to 6 experimental units.
Please provide a better explanation of the experimental design
Authors reply
- We corrected the information
Line 267 We change Thirty to Sixty
Line-272 Ten experimental units with (we change five to three) insects per treatment (we add thirty insects total) were evaluated, per treatment.
We tested 2 treatments. Treatment 1 control (30 insects) and treatment 2 caffeine emulsion treatment (30 insects).
- line 283-286 Here, authors must test differences in survival using Survival Analysis (e.g. Kaplan-Meier method)
Authors reply
As in shown in Figure 9. We believe that is not necessary to test this kind of differences, the effect on the insects were quite evident control 0% insect mortality vs Treatment 100% insect mortality. The survival analysis (e.g. Kaplan-Meier method) is used to estimate the probability that mortality occurs over time. In this the important variable is the time until the mortality occurs. In our case the respond variable is not the time but the mortality % that happen in a specific time. We really wanted to know the effect of the treatment on mortality
- line Why did authors report the acronyms of caffeine oleate emulsion with added kaolin in this part of the manuscript? Please correct the text reporting the acronyms when the caffeine oleate emulsion with added kaolin was reported for the first time
Authors reply
The information of Kaolin was added in the introduction because it was not in the original manuscript. In this revised version the kaolin information was added in lines 115-121 (it is marked in yellow)
With respect to the use of the acronyms in the table and figure before Figure 11, for a better undertanding we talk about emulsion and emulsion -kaolin. However, in the experiment 2.8 Evaluation of anatomical changes of CBB in contact with caffeine oleate by using scanning electron microscopy the insecticidal product was already tested and developed – we were not comparing treatment so we used the acronym.
- Line 307 and 310- What does "high" mean? Authors must report data as "The emulsions with and without kaolin caused XX and XX% mortality, respectively
Authors reply
We add the information 84% and 94% respectively.
- Line 309. Authors must report F value and degrees of freedom from ANOVA. These statistics are necessary to evaluate the correctness of the analysis as well.
Authors reply
We add all the ANOVA information Line 309 ANOVA, F= 453.48, DF= 2.24, P <0.0001). for % Mortality
- Line 313. Authors must report F value and degrees of freedom from ANOVA. These statistics are necessary to evaluate the correctness of the analysis as well.
Authors reply
We add all the ANOVA information. Line 313 (ANOVA, F= 76.28, DF=2.24, P<0.0001) for healthy seed.
- Line 315. After seeing this table I believe that analysis should be conducted in a different manner. In this case, the mortality observed in "water" treatment should be used to "correct" the mortality observed in aother treatment. In this manner, authors could estimate the mortality due to the tsted applications. Authors can use Schneider-Orelli formula if mortality is expressed as a percentage. After correcting the mortality, either a t-test or a Kolmogorov-Smirnov test must be conducted to test for differences between different applications
Authors reply
For us is important to know the natural mortality because is the mortality of the population that we are going to find in lab and field, and in our case it is not necessary to do corrections. The mortality causes by the emulsion is 84% and emulsion with kaolin is 94%, considering the natural mortality that we report in the control that is around 10%.
The two analyzes proposed by the reviewer would lead us to obtain the same results with respect to statistical differences.
Kolmogorov-Smirnov test m- is a test that compares two cumulative probability distributions. However, we were not interested in comparing survival curves. Our interest was not to determine the survival probabilities at each time, only to see the differences in mortality.
- Line 331. Here and here after, authors must report statistics (e.g., F value)
Authors reply
We add all the ANOVA information.
Line 334 (ANOVA, F=947.34, DF=2.42, P < 0.0001).
Line 340 (ANOVA, F=502.54, DF=2.42, P <0.0001).
Line 355 (ANOVA, F=498.78, DF=4.20, P <0.0001).
Line 361 (ANOVA, F=84.1, DF=4.20, P<0.0001)
Line 379 (ANOVA, F=44.28, DF=4.10, P < 0.0001).
Line 384 (ANOVA, F=10.78, DF=4.10, P = 0.0012).
Line 459 (ANOVA, F=150.71, DF=1.16, P<0.0001)
23 Line 367. In all the figures, I suggest to report letters above the bars to indicate separation among different groups
Authors reply
To have the information of figure and table separated makes easier to understand the results, for example in our figure 6, it shows the behavior of the data- variable % mortality over time (3 points). Meanwhile in the table 6 shows the statistical differences in the data- but only in one specific point at 20 days -different letters indicate differences according to Tukey 5%.
24.Line 506. How do the authors explain the differences in the effect of the tested emulsion between the two pests?
Authors reply
In both pest the effect takes between 2 to 5 days. However, some differences can be due to the differences in insect cuticle components and morphology. However, we only perform the microscopy study in CBB because of that we did not discuss this point deeper.
- Linea 491. Line In italics
Authors reply
The change M. velezangeli was done.
